# Correlation Between Prognostic Nutritional Index, Glasgow Prognostic Score, and Different Obesity-Related Indices in People with Diabetes or Prediabetes

**DOI:** 10.3390/diagnostics14232661

**Published:** 2024-11-26

**Authors:** Roxana-Viorela Ahrițculesei, Lidia Boldeanu, Ionela Mihaela Vladu, Diana Clenciu, Adina Mitrea, Radu Cristian Cîmpeanu, Maria-Lorena Mustață, Isabela Siloși, Mihail Virgil Boldeanu, Cristin Constantin Vere

**Affiliations:** 1Doctoral School, University of Medicine and Pharmacy of Craiova, 200349 Craiova, Romania; roxana.blendea@gmail.com (R.-V.A.); cimpeanu_r@yahoo.com (R.C.C.); umlorena@yahoo.com (M.-L.M.); 2Department of Microbiology, Faculty of Medicine, University of Medicine and Pharmacy of Craiova, 200349 Craiova, Romania; lidia.boldeanu@umfcv.ro; 3Department of Diabetes, Nutrition and Metabolic Diseases, Faculty of Medicine, University of Medicine and Pharmacy of Craiova, 200349 Craiova, Romania; ionela.vladu@umfcv.ro (I.M.V.); dianaclenciu@yahoo.com (D.C.); ada_mitrea@yahoo.com (A.M.); 4Department of Immunology, Faculty of Medicine, University of Medicine and Pharmacy of Craiova, 200349 Craiova, Romania; isabela_silosi@yahoo.com; 5Department of Gastroenterology, University of Medicine and Pharmacy of Craiova, 200349 Craiova, Romania; vere_cristin@yahoo.com

**Keywords:** prognostic nutritional index, Glasgow Prognostic Score, body mass index, waist to hip ratio, waist to height ratio, body adiposity index

## Abstract

Background/Objectives: The prognostic nutritional index (PNI) and Glasgow Prognostic Score (GPS) are associated with patients’ nutritional and immune statuses. One important factor in the pathophysiology of type 2 diabetes mellitus (T2DM) is inflammation. Being present in insulin-target tissues, chronic tissue inflammation has become recognized as a crucial aspect of obesity and type 2 diabetes. This study aimed to compare the PNI and GPS levels of the subjects with T2DM to those of prediabetes (preDM) individuals. Furthermore, the goal was to investigate how these inflammatory markers relate to different types of obesity and whether the combination of PNI, GPS, and obesity-related indices was associated with any particular prognostic variables. Methods: In this study, we enrolled one-hundred patients with newly diagnosed T2DM and one-hundred patients with preDM. Results: Four findings emerged from this observational study. As a first observation, 28% of patients with preDM and 15% of patients with T2DM had a normal weight, while up to 43% of patients with preDM and 60% of patients with T2DM were obese. The second important observation was that the PNI of the T2DM patients was significantly lower than the PNI of the patients with preDM (*p* < 0.0001). The PNI showed that patients with T2DM had a moderate-to-severe malnutrition status (median value of 38.00). Patients with preDM had a mild-to-moderate malnutrition status (median value of 61.00) at diagnosis. Third, observed in the current study, preDM patients with PNI < 61.00 and T2DM patients with a PNI < 38.00 were associated with significantly higher median values of the waist-to-height ratio (WHtR) (*p* = 0.041, and *p* = 0.034, respectively) and body mass index (BMI) (*p* = 0.016, and *p* = 0.041, respectively). Fourth, this study also revealed, in the T2DM group, a moderate and statistically significant negative correlation between PNI and weight (*rho* = −0.322, *p* = 0.035), waist circumference (WC) (*rho* = −0.308, *p* = 0.042), hip circumference (HC) (*rho* = −0.338, *p* = 0.039), WHtR (*rho* = −0.341, *p* = 0.022), body adiposity index (BAI) (*rho* = −0.312, *p* = 0.032), and fasting plasma glucose (FPG) (*rho* = −0.318, *p* = 0.029). Additionally, the PNI values expressed a weak negative correlation with BMI (*rho* = −0.279, *p* = 0.015), and glycated hemoglobin A1c (HbA1c) (*rho* = −0.245, *p* = 0.025). The PNI levels exhibited a single positive correlation, weak but statistically significant, with estimated glomerular filtration rate (eGFR-CKD-EPI) values (*rho* = 0.263, *p* = 0.018). Conclusions: The findings of this study regarding the correlations between PNI, GPS, and different obesity-related indices in people with diabetes or prediabetes suggest that these indices, which assess nutritional and inflammatory status, can be used as independent predictor factors associated with the four pillars of DM management (glucose, blood pressure, lipids, and weight control) recommended by the American Diabetes Association (ADA).

## 1. Introduction

Diabetes is one of the twenty-first century’s worldwide health emergencies with the quickest growth rate, according to the findings of the 10th edition of the International Diabetes Federation (IDF) Diabetes Atlas [1]. Forecasts indicate that 537 million people worldwide had diabetes in 2021, which will rise to 643 million by 2030, and reach 783 million by 2045. Furthermore, it is projected that 541 million individuals will have impaired glucose tolerance by 2021 [2].

Prediabetes (preDM) is a condition preceding diabetes wherein blood glucose is higher than normal yet below the diabetes limit [3,4]. It is typically defined as an intermediate state of hyperglycemic concentration in the blood with a high potential to progress to type 2 diabetes mellitus (T2DM) [4,5].

According to the American Diabetes Association (ADA), prediabetes is diagnosed when the fasting plasma glucose (FPG) is between 100 and 125 mg/dL or glycated hemoglobin A1c (HbA1c) levels are between 5.7 and 6.4% [3,4,5]. The progression from prediabetes to diabetes can occur slowly over the years; once the disease is established, it will be irreversible [6,7]. Therefore, awareness of the condition and immediate intervention can be indispensable to prevent or at least delay the onset of T2DM [8].

Around 5–10% of all prediabetic subjects develop T2DM yearly in the United States [9,10]. Some studies showed a conversion to diabetes can occur within five years if prediabetes is left untreated [11].

Studies have confirmed a strong association between obesity and prediabetes [12,13]. Indeed, central (visceral) obesity is strongly linked to developing T2DM [13]. Other risk factors include physical inactivity, hypertension, dyslipidemia (high triglycerides or low high-density lipoprotein, cholesterol), family history of diabetes, gestational diabetes, and smoking [12,13,14].

Adults with body mass index (BMI) ≥ 25 kg/m^2^ and additional risk factors should be screened for prediabetes. If there are no risk factors, screening should not occur later than age 45. FPG, two-hour plasma glucose (2 h PG) after a 75 g oral glucose tolerance test, and HbA1c are all validated tests to diagnose prediabetes [12]. Previously, studies [15,16,17] have frequently employed BMI to measure weight, but it was unable to differentiate between patients who were abdominally or generally obese. Further research is needed to determine whether the waist-to-hip ratio (WHR) waist-to-height ratio (WHtR), and body adiposity index (BAI) are related to immunological and nutritional status.

Studies to explore the relationship between obesity and the nutritional and immunological status among Romanian people are also limited. Therefore, this retrospective study assessed the association of obesity-related indices with immunological and nutritional factors, such as the prognostic nutritional index (PNI) and Glasgow Prognostic Score (GPS), among T2DM and preDM patients using the data of two university clinical hospitals representative for Dolj County, Romania. We also wanted to identify the possible correlation between them.

PNI is evaluated by the lymphocyte (LYM) count of the peripheral blood and serum albumin (ALB), which is a biomarker that integrates nutritional status, immune state, and inflammation condition. It was originally founded by Japanese scholars Onodera et al. [18,19,20,21] in 1984 to evaluate the nutritional and immunological condition of cancer patients having gastrointestinal surgery. In recent years, PNI has increasingly emerged as a novel prognostic indicator for various disorders, including heart failure [22,23,24,25] and the coronavirus disease 2019 (COVID-19) [26]. Diabetes and obesity have recently been identified as risk factors for severe COVID-19 disease [27,28]. The PNI has been identified as a predictor of mortality in older patients with chronic kidney disease (CKD) [29]. The correlation between PNI and the prognosis of diabetic nephropathy (DN) in patients with T2DM remains ambiguous. Furthermore, a notable correlation existed between serum ALB levels and the severity of retinopathy in T2DM [30].

## 2. Materials and Methods

Over six months, we carried out an epidemiological, non-interventional, and cross-sectional study. In this study, one-hundred-eighty-five consecutive patients with newly diagnosed T2DM were enrolled, while one-hundred patients with preDM who matched the inclusion criteria in terms of age, gender ratio, and urban/rural location made up the control group. This study was conducted by the Declaration of Helsinki, and approved by the Ethics Committee of the Filantropia Municipal Clinical Hospital (no. 886/15 January 2024) and Emergency County Clinical Hospital of Craiova (no. 2371/14 January 2022), Dolj, Romania.

### 2.1. Patient Selection

The following conditions had to be met to be included in the study: individuals with type 2 diabetes who were older than eighteen years were chosen from the Outpatient Diabetes, Nutrition, and Metabolic Diseases Departments of the Filantropia Municipal Clinical Hospital and the Emergency County Clinical Hospital of Craiova. All the participants were voluntarily included in the study, after signing the informed consent.

Patients with chronic microvascular complications of T2DM at diagnosis, which include diabetic peripheral polyneuropathy, diabetic kidney disease, and diabetic retinopathy were excluded from the study. Diabetic retinopathy (DR) was diagnosed following a dilated fundus examination [31]. As advised by the American Diabetes Association (ADA), diabetic peripheral neuropathy was evaluated using a combination of temperature sensation (for small fiber function) and vibration sensation (for large fiber function) tests, as well as the presence of characteristic symptoms (pain, dysesthesias, numbness) [31]. Guidelines from Kidney Disease: Improving Global Outcomes (KDIGO) were used to assess the existence of CKD [32].

Patients under the age of 18, pregnant women, those who had experienced an acute infection or inflammatory disease in the last month, those with a chronic infection or inflammatory disease, and cancer patients were excluded from the study.

### 2.2. Assessment of Diabetes and Prediabetes

One of the following criteria can be used to define prediabetes: (1) a diagnosis made by a medical professional; (2) a hemoglobin A1c (HbA1c) level greater than 5.7% and less than 6.5%; (3) a fasting plasma glucose (FPG) level between 5.6 mmol/L and 7.0 mmol/L; or (4) a 2 h FPG value during an oral glucose tolerance test (OGTT) between 7.8 mmol/L and 11.0 mmol/L [33]. Patients who presented with obesity (especially abdominal or visceral obesity), dyslipidemia with elevated triglycerides and/or low HDL cholesterol, and hypertension, and met the criteria mentioned above, were included in the preDM group.

A diagnosis of diabetes is made if one or more of the following conditions are satisfied: a medical diagnosis that has been verified by the patient’s healthcare providers; an HbA1c level that is greater than 6.5%; an FPG level of 7.0 mmol/L or higher; a random blood glucose level of 11.1 mmol/L or higher; a two-hour blood glucose level that is greater than 11.1 mmol/L after an OGTT; or random glucose value accompanied by classic hyperglycemic symptoms (e.g., polyuria, polydipsia, and unexplained weight loss) or hyperglycemic crises [33].

One-hundred out of one-hundred-eighty-five patients T2DM patients completed the study and were included in the final analysis, while eighty-five were lost to follow-up for the following reasons: diabetic peripheral polyneuropathy (*n* = 30), diabetic kidney disease (*n* = 25), diabetic retinopathy (*n* = 20), unwillingness to continue (*n* = 5), and relocation (*n* = 5).

### 2.3. Medical History, Biometric Parameter Assessment, and Demographic Data

Data on anthropometric measures, medical variables, laboratory test results, and demographic and lifestyle details were all intended to be gathered through an interview questionnaire.

Age, sex, household income each month, and educational attainment were among the demographic factors. Factors related to lifestyle and health included the presence of a smoking history or drinking history; a family history of hypertension, diabetes mellitus, and cardiovascular diseases; and the amount of time spent engaging in intentional moderate physical activity each week.

### 2.4. Different Obesity-Related Indices (BMI, WHR, WHtR, and BAI) Assessment

We determined the body mass index (BMI) using the participant’s height and weight measurements. The calculation is BMI = weight (kilograms)/height^2^ (meters). The patient’s nutritional status was evaluated according to BMI, using the WHO criteria [28]. BMI was categorized into normal weight (18.5–22.9 kg/m^2^), overweight (23.0–25.0 kg/m^2^), and obese (>25.0 kg/m^2^), according to the WHO. We measured the weight with a weight scale. Using a measuring stick on the weight scale, the height was determined.

The hip circumference (HC) over the femoral trochanters and the waist circumference (WC) at the halfway between the upper iliac crest and the lower border of the rib cage were measured. The waist-to-hip ratio (WHR), which is computed using the formula WC (cm)/HC (cm), was another tool used to measure abdominal obesity. Visceral adiposity was also evaluated using the waist-to-height ratio (WHtR), which was computed using the formula WC (cm)/height (cm). The body adiposity index (BAI) was calculated: BAI = ((hip circumference)/((height)1.5) − 18) [34]. Due to the lack of standard categories, we classified WHR, WHtR, and BAI into quarters.

### 2.5. Laboratory Investigations

After collecting anthropometric data, we brought the subjects to the laboratory for further investigation.

Laboratory data, blood urea nitrogen (BUN), creatinine (CREA), uric acid (UA), fasting plasma glucose (FPG), two-hour plasma glucose after a 75 g oral glucose tolerance test (2hPG), glycosylated hemoglobin A1c (HbA1c), total cholesterol (TC), total triglycerides (TG), low-density lipoprotein cholesterol (LDL-C), high-density lipoprotein cholesterol (HDL-C), C-reactive protein (CRP), and albumin (ALB) were determined using the chemiluminescence immunological technique and an automatic immunoassay analyzer (Cobas e411, Roche Diagnostics GmbH, Mannheim, Germany).

Using flow cytometry and Coulter’s principle, we were able to obtain an extended leukocyte formula of 5 diff (Ruby Cell-Dyne, Abbott, Abbott Park, IL, USA) and determine the hemoleucogram markers: hemoglobin (Hb), white blood cells/leukocytes (WBC), neutrophils (NEU), lymphocytes (LYM), monocytes (MON), platelets (PLT), and hemoglobin (Hb).

Measurements of serum creatinine were made, and the Chronic Kidney Disease Epidemiology Collaboration (CKD-EPI) formula [35], and the Modification of Diet in Renal Disease Study (MDRD-Study) [36] were used to determine the estimated glomerular filtration rate (eGFR).

### 2.6. Prognostic Nutritional Index and Glasgow Prognostic Score Calculations

Based on the absolute lymphocyte count and serum ALB level, the prognosis nutritional index (PNI) is calculated. The PNI was calculated according to the acknowledged formula: 10 × serum albumin (g/dL) + 0.5% × total lymphocyte number (per mm^3^) [37]. Interpretation: PNI value ≥ 50—Normal, PNI value < 50—Mild malnutrition, PNI value < 45—Moderate-to-severe malnutrition, PNI value < 40—Serious malnutrition.

CRP and ALB levels were used to calculate the Glasgow Prognostic Score (GPS); patients with CRP ≤ 10 mg/L and ALB ≥ 35 g/L were allocated to the GPS0 group. Patients with only CRP > 10 mg/L were assigned to the GPS1 group. Patients who had both CRP > 10 mg/L and ALB < 35 g/L [38] were assigned to the GPS2 group.

Provided that 61.00 was the median value among the 100 preDM patients, and 38.00 for the 100 T2DM patients, respectively, we used the median of PNI scores as classified criteria and patients were divided into two groups: low PNI (<61.00, and <38.00, respectively) group and high PNI (≥61.00, and ≥38.00, respectively) group.

### 2.7. Statistical Analysis

Using Microsoft Excel, we processed and handled patient data from medical records. To analyze the data, we utilized GraphPad Prism 10.3.1 Version (GraphPad Software, LLC, San Diego, CA, USA). The D’Agostino and Pearson normality tests were used to determine whether the data were normal.

The following variables’ means are shown alongside their standard deviations (SD): Hb, WBC, NEU, LYM, MON, PLT, ALB, CRP, ESR, BUN, Crea, UA, FPG, 2hPG, HbA1c, TC, TG, LDL-C, HDL-C, and BUN all had normal distributions. It was demonstrated that the distributions of height, weight, WC, HC, WHR, WHtR, and BAI were non-normal, and the data are displayed as the median with interquartile range. The category values are stated as percentages.

Continuous variables were evaluated using the one-way ANOVA or the Kruskal–Wallis test (used for non-Gaussian distributions) to find the difference between groups and the *χ*^2^ test was used for categorical variables.

Spearman’s coefficients (−1 < *rho* < 1) were used to see if there were any significant correlations between the levels of BMI, height, weight, WC, HC, WHR, WHtR, BAI, ALB, CRP, WBC, NEU, LYM, MON, PLT, PNI, and ESR.

## 3. Results

### 3.1. Clinical and Demographic Features of the Patients with Prediabetes and Diabetes

In this study, we included 100 patients diagnosed with T2DM, aged between 31 and 75 years, with a mean ± standard deviation (SD) age of 54.11 ± 6.33, consisting of 43 women and 57 men. In the control group, the preDM group, we found that the mean ± SD age was 50.08 ± 7.46, and women predominated with a percentage of 56. Thus, a statistically significant difference was observed regarding age (*p* = 0.003), but not in the case of gender (*χ*^2^(1) = 3.38, *p* = 0.066). In terms of where the patients lived, we saw that most of the people in both the T2DM and preDM groups (65 and 61 patients, respectively) were from rural areas. There was no statistically significant difference between the two groups (*χ*^2^(1) = 0.34, *p* = 0.560).

Analyzing the lifestyle factors of smoking and drinking histories, we identified that more patients with T2DM are smokers and alcohol consumers, with the differences being statistically significant (*p* = 0.007 and, respectively, *p* < 0.0001).

Personal history was another important difference between the T2DM and the preDM group based on statistics; diseases such as having hypertension, dyslipidemia, or hepatosteatosis were found in more than 55% of patients. In the same context, systolic blood pressure (SBP), and diastolic blood pressure (DBP) had statistically significantly higher mean values in the T2DM group.

Regarding the measured anthropometric parameters (height, weight, waist circumference (WC), and hip circumference (HC)) and different obesity-related indices (WHR, WHtR, BAI, and BMI), there were statistically significant differences between the two groups for the mean values of height (*p* = 0.009) and the medians for WHR (*p* = 0.018), while the mean values of weight and BMI reached the significance limit (*p* = 0.059 and *p* = 0.055, respectively).

The analysis of laboratory parameters highlighted that both the parameters determined for diagnosing T2DM and those of the lipid profile showed statistically significant higher mean values compared to the means of the preDM group, as can be seen in Table 1.

We looked at nutritional status and systemic inflammation in our study. We discovered that levels of the CRP (47.91 ± 36.27 vs. 27.46 ± 19.65, *p* < 0.0001), ALB (3.89 ± 0.78 vs. 6.19 ± 0.37, *p* < 0.0001), white blood cells (WBC) (8.68 ± 2.83 vs. 7.92 ± 1.93, *p* = 0.026), and LYM (2.67 ± 0.89 vs. 2.27 ± 0.71, *p* < 0.0001) were significantly different between the two groups (T2DM vs. preDM). PNI determined by the number of LYM in peripheral blood and serum ALB showed statistically significantly different values between the preDM and T2DM group, as well as the fact that patients with T2DM had a moderate-to-severe malnutrition status at diagnosis. Based on the GPS value, which reflects both the inflammatory and nutritional status using ALB and CRP levels, we found that 39% of T2DM patients had moderate-to-severe malnutrition status. This was in contrast to 79% of preDM patients who had mild-to-moderate malnutrition status.

### 3.2. Comparing the PNI and GPS Groups’ Clinical Features Between the preDM and T2DM Groups

Utilizing PNI cut-off values of 61.0 and 38.0, we categorized both groups into two subgroups: low PNI (<61.00 and <38.00, respectively) and high PNI (≥61.00 and ≥38.00, respectively) (Table 2). Additionally, the GPS data were categorized into two subgroups: GPS1 as subgroup 1, and GPS2 as subgroup 2, in the T2DM group, and GPS0 as subgroup 1, and GPS1 as subgroup 2, in the preDM, respectively.

The majority of patients (67%) in the preDM group had a PNI ≥ 61.00 and a GPS2 score (79 patients). Additionally, in the T2DM group, just over 50 percent of patients had a PNI ≥ 38.00, but 61% of them recorded a GPS1 score.

There were no differences in age, gender, and area of residence (*p* ≥ 0.05), both in the preDM subgroups (PNI < 61.00 and PNI ≥ 61.00, and GPS1 and GPS2, respectively) and in the T2DM subgroups (PNI < 38.00 and PNI ≥ 38.00, and GPS1 and GPS2, respectively).

In the preDM group, we saw that PNI values below 61.00, which means a moderate nutritional and inflammatory status, were strongly linked (*p* < 0.0001) to lifestyle factors like smoking and drinking, education, and the main risk factors that can lead to T2DM, hypertension, dyslipidemia, and hepatosteatosis. Additionally, in the preDM group, over 38% of patients with a GPS2 score presented smoking, hypertension, dyslipidemia, and hepatosteatosis as risk factors. Thus, we can consider that these risk factors can influence and induce a moderate-to-severe nutritional and inflammatory status. This nutritional and inflammatory status is also supported by ALB, which presented significantly lower mean values (*p* < 0.0001). In the PNI < 61.00 subgroups, the patient’s mean values of WBC (*p* = 0.038) and neutrophils (NEU) (*p* = 0.018) were significantly different compared to those in the PNI ≥ 61.00 subgroup.

Analyzing the association between the PNI index and different obesity-related indices, it was observed that patients in the PNI < 61.00 subgroups showed significant changes only for the median values of weight (*p* = 0.038), WC (*p* = 0.032), HC (*p* = 0.022), WHtR (*p* = 0.041), and BMI (*p* = 0.016) compared to those in the PNI ≥ 61.00 subgroups.

In the T2D group, we observed that lifestyle factors (smoking and drinking), education, and the main risk factors of T2DM (hypertension, dyslipidemia, hepatosteatosis, and SBP) were present in more than 54.54% of patients with PNI < 38.00, or patients with a GPS2 score.

Anthropometric parameters, such as weight, WC (*p* = 0.023), HC (*p* = 0.035), and obesity-related indices, WHtR (*p* = 0.034) and BMI (*p* = 0.041), presented significantly higher median values in the PNI < 38.00 subgroups, and the WHR index (*p* = 0.055) and BAI (*p* = 0.057), reached the significance limits, compared to those in the PNI ≥ 38.00 subgroup.

Among patients diagnosed with T2DM, those in the PNI < 38.00 subgroups had a moderate-to-severe nutritional and inflammatory status, associating significantly modified WBC, NEU, and ALB mean values comparable to those in the PNI ≥ 38.00 subgroups.

Patients in PNI < 38.00 subgroups had mean values of fasting plasma glucose (FPG) (*p* = 0.049), HbA1c (*p* = 0.032), total cholesterol (TC) (*p* = 0.015), and low-density lipoprotein cholesterol (LDL-C) (*p* = 0.021) significantly different from those in the PNI ≥ 38.00 subgroups. Also, patients who had a GPS2 score were associated with higher values of HbA1c and TC.

### 3.3. Associations of PNI with BMI, WHR, WHtR, and BAI in the preDM and T2DM Groups

Table 3 and Figure 1 display how PNI and BMI, WHR, WHtR, and BAI relate to each other in the preDM and T2DM groups. BMI was categorized into normal weight (18.5–22.9 kg/m^2^), overweight (23.0–25.0 kg/m^2^), and obese (>25.0 kg/m^2^), according to the World Health Organization (WHO) [39]. Due to the lack of standard categories, we classified WHR, WHtR, and BAI into quarters.

Using the one-way ANOVA test, we obtained that preDM patients had statistically significant differences in values between BMI categories (*p* = 0.019). On the other hand, the Kruskal–Wallis test highlighted that in the case of preDM patients, the differences between the quarters of WHR and BAI reached the significance limits (*p* = 0.059 and *p* = 0.053, respectively).

There were no statistically significant differences in values between BMI categories and differences between the quarters of WHR, WHtR, and BAI, respectively, in the T2DM group.

### 3.4. Correlations Between PNI and the Obesity-Related Indices in the preDM and T2DM Groups

Spearman’s correlation analysis revealed that the values of PNI correlated much better with the obesity-related indices in the T2DM group (Figure 2).

In the T2DM group, our study revealed a moderate and statistically significant negative correlation between PNI and weight (*rho* = −0.322, *p*-value = 0.035), WC (*rho* = −0.308, *p*-value = 0.042), HC (*rho* = −0.338, *p*-value = 0.039), WHtR (*rho* = −0.341, *p*-value = 0.022), BAI (*rho* = −0.312, *p*-value = 0.032), and FPG (*rho* = −0.318, *p*-value = 0.029). Additionally, the PNI values expressed a negative weak correlation with BMI (*rho* = −0.279, *p*-value = 0.015) and HbA1c (*rho* = −0.245, *p*-value = 0.025). We found a single positive correlation, weak but statistically significant, between PNI and CKD-EPI (*rho* = 0.263, *p*-value = 0.018).

## 4. Discussion

Globally, 26.6 million people are expected to have diabetes in 2025, whereas 579.9 million people will have the disease [40]. The WHO projects that diabetes’s rising prevalence will make it the seventh-largest cause of death globally by 2030 [41]. Furthermore, it is also expected that the number of people with preDM—a condition marked by blood glucose levels that are higher than normal but fall short of the diagnostic cut-off for T2DM—will increase [4].

Forecasts indicate that by 2030, there will be more than 470 million people living with preDM worldwide [42]. Importantly, one should not undervalue preDM. People with preDM are associated with an increased risk of cardiovascular disease and diabetic microangiopathy as compared to people with normal glucose metabolism [43,44].

Insulin resistance and malfunction of the pancreatic beta cells are important markers of DM and preDM. Given the varying effects of these factors on preDM subgroups across diverse racial and ethnic groups, it is imperative to establish prevention initiatives and effective treatment procedures for DM and preDM [45,46]. Remarkably, the proportion of people with preDM greatly exceeds that of people with DM [47]. Therefore, early diagnosis of preDM and DM is essential for effective care and the prevention of the disease’s progression [48,49].

In this study, we enrolled one-hundred patients with newly diagnosed T2DM and one-hundred patients with preDM and analyzed the association of obesity-related indices with the immunological and nutritional factors, PNI, and GPS, and identified the possible correlation between them.

From our observational study, four findings emerged. As a first observation, 28% of patients with preDM and 15% of patients with T2DM had a normal weight, while up to 43% of patients with preDM and 60% of patients with T2DM were obese. The second important observation was that the PNI of the T2DM patients was significantly lower than the PNI of the patients with preDM. The PNI showed that patients with T2DM had a moderate-to-severe malnutrition status (median value of 38.00). Patients with preDM had a mild-to-moderate malnutrition status (median value of 61.00) at diagnosis. Third, we observed that preDM patients with PNI < 61.00 and T2DM patients with PNI < 38.00 were associated with significantly higher median values of WHtR and BMI. Fourth, in the T2DM group, our study revealed a moderate and statistically significant negative correlation between PNI and weight, WC, HC, WHtR, BAI, and FP. Additionally, the PNI values expressed a weak negative correlation with BMI and HbA1c. The PNI levels exhibited a single positive correlation, weak but statistically significant, with CKD-EPI values.

Obesity and type 2 diabetes have a lot of complicated connections and shared pathophysiological pathways. This makes it more likely for obese people to have insulin resistance, dyslipidemia, non-alcoholic fatty liver disease, and many other metabolic problems [50,51]. Due to changes in adipose tissue biology that link obesity with insulin resistance and beta cell dysfunction, an increased BMI and the distribution of fat around the abdomen linearly increase the risk of type 2 diabetes [52]. Obesity leads to chronic, systemic inflammation [53]. Like in endometriosis, the appearance of cellular atypia or malignant transformation can occur under the influence of the proinflammatory microenvironment, particularly in inflammatory cells, which provides a favorable environment for neovascularization and the presence of mutations in tumor suppressor proteins or oncoproteins, with an associated increase in cell proliferation and tumor growth [54].

To lower the frequency of complications, weight control is typically advised in the treatment of diabetes and several systemic disorders [55]. Even when the overweight person with a predominance of abdominal fat does not match the BMI requirements for obesity, abdominal obesity, which is frequently measured by indices such as the WHR, BAI, or visceral adiposity index, is an independent predictor for the development of hypertension and high fasting glucose [52,56]. The BMI is not a reliable indicator of adiposity; De Lorenzo has demonstrated that a more accurate measure could be the anthropometric measurement of body fat percentage [57]. A study published in 2020 in the United States reported that approximately two-thirds of the analyzed adult population was obese or overweight [58].

Referring to the patients included in our study, we observed the same findings: 25% of patients with T2DM were overweight and 60% were obese. The observation can be explained by the fact that Romania has quickly adopted a relatively high socioeconomic status, with an increase in income per capita, circumstances that have led to a higher prevalence of obesity.

Li et al. [34] found that the three obesity-related indices, BMI, WHtR, and BAI, were negatively associated with the development of diabetic retinopathy (DR). According to the authors, there might be a lack of differentiation between general obesity and centripetal obesity using the BMI, and the two may have differential effects on the development of diabetes [57]. Regarding this aspect, Man et al. [59] found a positive correlation between centripetal obesity, indicated by a higher WHR, and the progression of diabetes. In another study, it was shown that WHR should be considered to assess centripetal obesity, while it was established that BAI has a significant linear relationship with body fat percentage [60]. They have also shown that abdominal obesity can be a more critical factor in diabetic retinopathy than generalized obesity.

In our study, we observed patients with T2DM-associated centripetal obesity, indicated by elevated values of WHR and WHtR. Additionally, nearly half of these patients, who had a moderate-to-severe nutritional and inflammatory status, were significantly associated with centripetal obesity. In contrast, only 33% of the preDM group exhibited a moderate-to-severe nutritional and inflammatory status, which was also associated with centripetal obesity.

As far as we know, there have been no population-based studies reporting the association between WHR, WHtR, or BAI measures and PNI index in patients with newly diagnosed T2DM or preDM.

Recent investigations have established the role of PNI as an emerging clinical marker of diagnostic and prognostic significance in patients with diabetes who present microvascular complications [61,62,63,64,65,66].

Aktas et al. discovered that the PNI of the type 2 diabetes mellitus patients was significantly lower than the PNI of the healthy controls [61]. Additionally, they noticed that the PNI of the diabetic subjects with microvascular complications (diabetic nephropathy, diabetic retinopathy, and diabetic neuropathy) was considerably lower than the PNI of the healthy controls and the diabetic patients without microvascular complications.

We obtained similar results in our study with newly diagnosed patients with T2DM. The PNI of the T2DM patients was significantly lower than the PNI of the patients with preDM, and the PNI showed that patients with T2DM had a moderate-to-severe malnutrition status, and patients with preDM had a mild-to-moderate malnutrition status at diagnosis.

Zhang et al. hypothesized that the PNI, which reflects nutrition, immunology, and inflammation—all of which are closely related to diabetic nephropathy (DN)—may be a more reliable predictor of end-stage renal disease (ESRD) in patients with DN than serum albumin, the inflammation index, or the lymphocyte count [62]. The same author found in another study that there was a significant correlation between all participants’ greater PNI and their risk of mortality and prevalence of CKD [63].

Kurtul et al. [64] found an independent and negative correlation between PNI and the presence of diabetic retinopathy (DR). Using PNI to examine a patient’s inflammatory–nutritional balance may help doctors identify T2DM patients who are at heightened risk for developing DR. According to Wei et al. [66], patients with DR also tended to be malnourished compared to non-DR patients, and malnourished patients had a greater incidence of DR than normal nutritional status patients; there was an independent correlation between the intensity and existence of DR and malnutrition. Yang et al. observed a strong correlation between a lower incidence of DR and greater PNI levels in hospitalized T2DM patients. The severity and prevalence of DR were found to be inversely and independently correlated with PNI, indicating that PNI may be utilized to predict the prognosis of DR in clinical practice [65].

Our study also revealed, in the T2DM group, a moderate and statistically significant negative correlation between PNI and weight, WC, HC, WHtR, BAI, and FPG. Additionally, the PNI values expressed a weak negative correlation with BMI and HbA1c. The PNI levels exhibited a single positive correlation, weak but statistically significant, with CKD-EPI values. Our findings were in agreement with those mentioned in other studies. PNI was found to be positively connected with eGFR and negatively correlated with BMI, HbA1c, and fasting blood glucose, according to Aktas et al. [61]. Additionally, Zhang et al. [62,63] found that the PNI was positively correlated with eGFR and discovered a negative correlation between the PNI and the following: red cell distribution width, HbA1c, urine albumin-to-creatinine ratio, neutrophil-to-lymphocyte ratio, glomerular injury, and high-sensitivity C-reactive protein.

The PNI is made up of serum albumin and blood lymphocyte count, both of which are easily obtained in the laboratory of practically every medical facility. The PNI is therefore a marker that is simple to evaluate and is a cheap marker since it is economical to study serum albumin and lymphocyte count.

Since this study was restricted to two university clinical hospitals representative of Dolj County, we acknowledge that it has inherent limitations. In addition, as we lacked a pilot study and prior data that were referenced in the literature, we were unable to determine the effect size and hence did not compute the simulation of sample size.

Patients with serious complications, a low willingness to seek medical attention, or mobility issues would be few, which would bias the study’s selection process. However, the limited number of patients in certain categories following their grouping might have an impact on the outcomes.

Therefore, more studies should be designed to investigate the association of obesity-related indices with the PNI, regarding different categories of patients with diabetes and prediabetes.

## 5. Conclusions

From our observational study, four conclusions emerged. As a first observation, up to 43% of patients with preDM and 60% of patients with T2DM were obese. The PNI showed that patients with T2DM had a moderate-to-severe malnutrition status, and patients with preDM had a mild-to-moderate malnutrition status at diagnosis. We observed that preDM and T2DM patients with low PNI were significantly associated with higher median values of WHtR and BMI. In the T2DM group, our study revealed a moderate and statistically significant negative correlation between PNI and weight, WC, HC, WHtR, BAI, and FP. Additionally, the PNI values expressed a weak negative correlation with BMI and HbA1c. The PNI levels exhibited a single positive correlation, weak but statistically significant, with CKD-EPI values. Our results can undoubtedly serve as a springboard for additional research and multicentric, long-term investigations. The findings of this study regarding the correlations between PNI, GPS, and different obesity-related indices in people with diabetes or prediabetes suggest that these indices, which assess nutritional and inflammatory status, can be used as independent predictor factors associated with the four pillars of DM management (glucose, blood pressure, lipids, and weight control) recommended by the ADA.

## Figures and Tables

**Figure 1 diagnostics-14-02661-f001:**
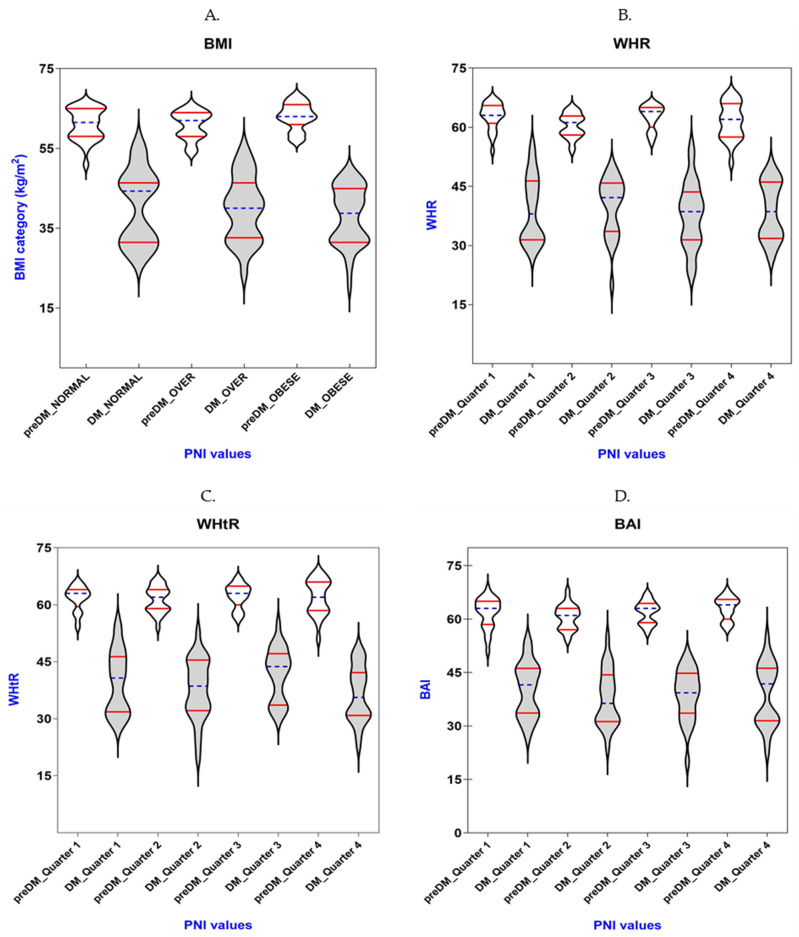
The PNI levels for patients with prediabetes (white color) or diabetes (gray color) vary in different quarters of the obesity-related indices: (**A**) BMI; (**B**) WHR; (**C**) WHtR; (**D**) BAI. Violin plot represents values of indices; horizontal blue lines represent median values accompanied by the quartiles represented by horizontal red lines.

**Figure 2 diagnostics-14-02661-f002:**
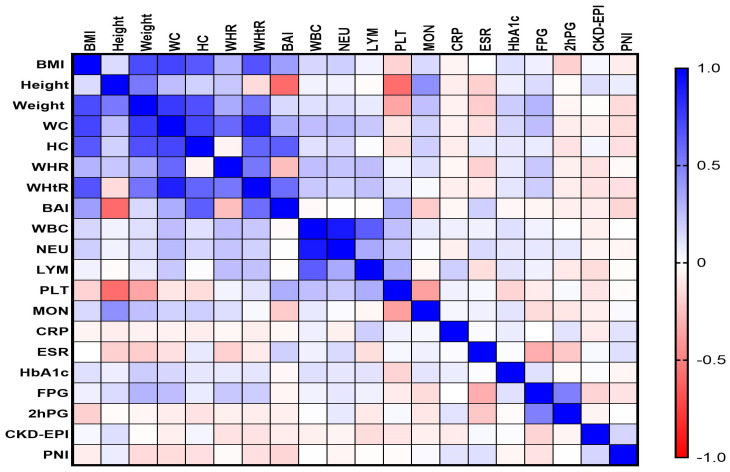
Correlations matrix between PNI and the obesity-related indices in the T2DM group. The correlation heatmap shows how the measured indicators relate to one another. Strong positive correlations are indicated by bright blue, whereas strong negative correlations are indicated by bright red.

**Table 1 diagnostics-14-02661-t001:** Clinical and demographic features of the patients with prediabetes and diabetes.

Characteristics	preDM Group (*n* = 100)	T2DM Group (*n* = 100)	*p*-Value from Pearson’s Chi-Squared/ Student’s *t*-Test
*Demographic characteristics*			
Age (yrs) (mean ± SD)	50.08 ± 7.46	54.11 ± 6.33	0.003 *
Gender, Female/Male (*n*)	56/44	43/57	0.066
Area of residence, Rural/Urban (*n*)	61/39	65/35	0.560
*Medical history and clinical condition*			
Smoking history, No/Yes (*n*)	57/43	38/62	0.007 *
Drinking history, No/Yes (*n*)	72	44/56	<0.0001 *
Education, No/Yes (*n*)	63/37	75/25	0.034 *
Hypertension, *n* (%)	47 (47%)	78 (78%)	<0.0001 *
Dyslipidemia, *n* (%)	43 (48%)	68 (68%)	0.0004 *
Hepatosteatosis, *n* (%)	34 (34%)	55 (55%)	0.003 *
SBP (mmHg) (mean ± SD)	132.20 ± 16.70	137.40 ± 15.67	0.018 *
DBP (mmHg) (mean ± SD)	77.81 ± 12.11	83.86 ± 10.65	0.0006 *
Height (cm) (mean ± SD)	1.64 ± 0.15	1.69 ± 0.11	0.009 *
Weight (kg) [median (range)]	83.00 (56.00–140.50)	89.00 (17.00–122.00)	0.059 **
WC (cm) [median (range)]	100.00 (58.00–171.00)	105.00 (80.00–143.00)	0.209
HC (cm) [median (range)]	106.00 (65.00–195.00)	106.50 (76.00–146.00)	0.263
WHR [median (range)]	0.94 (0.59–1.68)	0.98 (0.86–1.18)	0.018 *
WHtR [median (range)]	61.02 (33.92–95.00)	61.70 (44.20–87.50)	0.715
BMI (kg/m^2^) (mean ± SD)	28.18 ± 6.09	32.01 ± 5.24	0.055 **
BMI category (*n*)			
Normal (18.5–24.9 kg/m^2^)	28	15	0.031 *
Overweight (25–29.9 kg/m^2^)	29	25	0.256
Obese (≥30 kg/m^2^)	43	60	0.010 *
BAI [median (range)]	30.95 (10.08–67.70)	29.93 (16.29–62.27)	0.197
*Laboratory examination*			
FPG (mmol/L) (mean ± SD)	108.50 ± 6.12	210.20 ± 27.83	<0.0001 *
2hPG (mmol/L) (mean ± SD)	167.20 ± 14.14	332.30 ± 40.23	<0.0001 *
HbA1c (%) (mean ± SD)	5.82 ± 0.52	10.17 ± 1.98	<0.0001 *
TC (mmol/L) (mean ± SD)	183.30 ± 47.33	220.20 ± 54.43	<0.0001 *
TG (mmol/L) (mean ± SD)	125.50 ± 65.81	188.00 ± 108.60	<0.0001 *
LDL-C (mmol/L) (mean ± SD)	103.70 ± 42.13	136.50 ± 48.70	<0.0001 *
HDL-C (mmol/L) (mean ± SD)	53.64 ± 12.78	43.28 ± 13.38	<0.0001 *
eGFR (mL/min/1.73 m^2^)			
CKD-EPI [median (range)]	87.50 (38.00–117.00)	90.50 (29.00–129.00)	0.993
MDRD-STUDY [median (range)]	83.50 (38.00–147.00)	89.50 (29.00–147.00)	0
BUN (mg/dL) (mean ± SD)	41.62 ± 17.07	40.79 ± 14.81	0.718
Crea (mg/dL) (mean ± SD)	0.87 ± 0.37	0.91 ± 0.24	0.454
UA (mg/dL) (mean ± SD)	5.61 ± 1.46	4.98 ± 1.61	0.006 *
Hb (g/dL) (mean ± SD)	13.50 ± 1.86	14.55 ± 1.73	0.0002 *
WBC (×10^3^/μL) (mean ± SD)	7.92 ± 1.93	8.68 ± 2.83	0.026 *
NEU (×10^3^/μL) (mean ± SD)	4.87 ± 1.47	5.29 ± 3.02	0.211
LYM (×10^3^/μL) (mean ± SD)	2.27 ± 0.71	2.67 ± 0.89	0.0001 *
MON (×10^3^/μL) (mean ± SD)	0.53 ± 0.16	0.55 ± 0.18	0.298
PLT (×10^3^/μL) (mean ± SD)	248.30 ± 75.02	233.10 ± 71.48	0.177
*Malnutrition*			
ALB (g/dL) (mean ± SD)	6.19 ± 0.37	3.89 ± 0.78	<0.0001 *
CRP (mg/dL) (mean ± SD)	27.46 ± 19.65	47.91 ± 36.27	<0.0001 *
ESR (mm/1st h) (mean ± SD)	35.01 ± 25.42	38.43 ± 16.99	0.146
PNI	61.87 ± 3.67	38.87 ± 7.80	<0.0001 *
GPS, *n*			
0	21	-	-
1	79	61	-
2	-	39	-

SBP: systolic blood pressure; DBP: diastolic blood pressure; WC: waist circumference; HC: hip circumference; WHR: waist-to-hip ratio; WHtR: waist-to-height ratio; BMI: body mass index; BAI: body adiposity index; FPG: fasting plasma glucose; 2hPG: two-hour plasma glucose after a 75 g oral glucose tolerance test; HbA1c: glycosylated hemoglobin A1c; TC: total cholesterol; TG: total triglycerides; LDL-C: low-density lipoprotein cholesterol; HDL-C: high-density lipoprotein cholesterol; e-GFR: estimated glomerular filtration rate; CKD-EPI: chronic kidney disease epidemiology collaboration; MDRD-Study: modification of diet in renal disease study; BUN: blood urea nitrogen; CREA: creatinine; UA: uric acid; Hb: hemoglobin; WBC: white blood cells/leukocytes; NEU: neutrophils; LYM: lymphocytes; MON: monocytes; PLT: platelets; ALB: albumin; CRP: C-reactive protein; ESR: erythrocyte sedimentation rate; PNI, prognostic nutritional index; GPS: Glasgow Prognostic Score; SD: standard deviation; * *p* < 0.05: statistically significant; **: reached the significance limit.

**Table 2 diagnostics-14-02661-t002:** Comparing the PNI and GPS groups’ clinical features between the preDM and T2DM groups.

Characteristics		preDM Group (*n* = 100)						T2DM Group (*n* = 100)						
		PNI			GPS				PNI			GPS		
	All Patients	PNI < 61	PNI ≥ 61	*p*-Value	1	2	*p*-Value	All Patients	PNI < 38	PNI ≥ 38	*p*-Value	1	2	*p*-Value
Patients (*n*)	100	33	67		21	79		100	44	56		61	39	
*Demographic characteristics*														
Age (yrs) (mean ± SD)	50.08 ± 7.46	48.58 ± 7.46	50.82 ± 7.41	0.065	50.67 ± 8.24	49.92 ± 7.29	0.724	54.11 ± 6.33	52.91 ± 6.36	53.27 ± 6.67	0.814	53.21 ± 6.55	52.95 ± 6.05	0.619
Gender, Female/Male (*n*)	56/44	21/12	35/32	0.279	10//11	46/33	0.383	43/57	16/28	27/29	0.235	29/32	14/25	0.251
Area of residence, Rural/Urban (*n*)	61/39	20/13	41/26	0.862	12/9	49/30	0.680	65/35	25/19	40/16	0.128	42/19	23/16	0.312
*Medical history and clinical condition*														
Smoking history, Yes (*n*)	43	23	20	0.0001 *	13	30	0.049 *	62	35	27	0.001 *	23	39	<0.0001 *
Drinking history, Yes (*n*)	28	18	10	<0.0001 *	8	20	0.247	56	32	24	0.003 *	20	36	<0.0001 *
Education, No (*n*)	63	33	30	<0.0001 *	11	52	0.256	75	38	37	0.020 *	43	32	0.192
Hypertension, Yes (*n*)	47	29	18	<0.0001 *	5	42	0.017 *	78	24	54	<0.0001 *	58	20	<0.0001 *
Dyslipidemia, Yes (*n*)	43	24	19	<0.0001 *	5	38	0.046 *	68	24	44	0.011 *	48	20	0.004 *
Hepatosteatosis, Yes (*n*)	34	17	17	0.009 *	3	31	0.032 *	55	19	36	0.035 *	40	15	0.008 *
SBP (mmHg) (mean ± SD)	132.20 ± 16.70	130.88 ± 15.07	132.79 ± 17.53	0.306	133.10 ± 16.17	131.91 ± 16.94	0.891	137.40 ± 15.67	140.2 ± 15.99	135.2 ± 15.19	0.038 *	135.62 ± 14.80	140.10 ± 16.77	0.082
DBP (mmHg) (mean ± SD)	77.81 ± 12.11	79.09 ± 9.42	77.18 ± 13.26	0.661	78.19 ± 14.80	77.71 ± 11.40	0.909	83.86 ± 10.65	85.0 ± 8.86	82.96 ± 11.87	0.133	87.78 ± 11.61	85.54 ± 8.83	0.130
Height (m) (mean ± SD)	1.64 ± 0.15	1.64 ± 0.09	1.68 ± 0.10	0.441	1.69 ± 0.10	1.68 ± 0.08	0.664	1.69 ± 0.11	1.69 ± 0.08	1.68 ± 0.10	0.825	1.69 ± 0.10	1.68 ± 0.08	0.765
Weight (kg) [median (range)]	83.0 (56.0–140.5)	87.0 (56.0–140.5)	73.0 (56.0–101.6)	0.038 *	82.0 (56.0–140.5)	99.0 (60.0–132.0)	0.067	89.0 (17.0–122.0)	97.45 (17.0–122.0)	82.15 (53.0–115.0)	0.023 *	80.0 (53.0–115.0)	97.50 (17.0–122.0)	0.084
Waist circumference (cm) [median (range)]	100.0 (58.0–171.0)	108.0 (58.0–171.0)	92.0 (69.0–138.0)	0.032 *	93.0 (58.0–125.0)	106.0 (69.0–171.0)	0.102	105.0 (80.0–143.0)	115.10 (80.0–143.0)	94.75 (80.0–140.0)	0.029 *	101.50 (80.0–140.0)	109.50 (80.0–143.0)	0.076
Hip circumference (cm) [median (range)]	106.0 (65.0–195.0)	118.0 (75.0–133.0)	97.0 (65.0–195.0)	0.022 *	103.0 (65.0–147.0)	110.2 (86.0–195.0)	0.073	106.5 (76.0–146.0)	112.35 (76.0–146.0)	98.25 (87.0–130.0)	0.035 *	99.75 (76.0–130.0)	113.25 (87.0–146.0)	0.058 **
WHR [median (range)]	0.94 (0.59–1.68)	0.91 (0.59–1.64)	0.96 (0.64–1.68)	0.232	0.85 (0.59–1.11)	0.94 (0.77–1.68)	0.113	0.98 (0.86–1.18)	1.05 (0.86–1.18)	0.95 (0.88–1.18)	0.055 **	1.08 (0.88–1.18)	0.97 (0.86–1.18)	0.101
WHtR [median (range)]	61.02 (33.92–95.00)	64.84 (33.92–95.00)	56.35 (44.92–88.46)	0.041 *	58.49 (33.92–74.10)	61.35 (42.94–95.00)	0.187	61.7 (44.2–87.5)	67.3 (49.4–87.5)	56.7 (44.2–87.5)	0.034 *	60.50 (49.4–87.5)	65.15 (44.2–87.5)	0.069
BMI (kg/m^2^) (mean ± SD)	28.18 ± 6.09	32.54 ± 6.12	26.17 ± 5.79	0.016 *	28.5 ± 5.90	30.7 ± 6.81	0.082	32.01 ± 5.24	34.06 ± 3.44	29.41 ± 5.58	0.041 *	28.55 ± 3.15	35.05 ± 1.90	0.035 *
BMI category (*n*)														
Normal (18.5–24.9 kg/m^2^)	28	12	16		18	10		15	7	8		9	6	
Overweight (25–29.9 kg/m^2^)	29	12	17		13	16		25	9	16		16	9	
Obese (≥30 kg/m^2^)	43	9	34		27	16		60	28	32		37	23	
BAI [median (range)]	30.95 (10.08–67.70)	33.78 (14.12–66.23)	30.63 (15.84–44.60)	0.074	36.85 (24.37–66.23)	30.87 (14.12–48.93)	0.088	29.93 (16.29–62.27)	35.18 (21.36–62.27)	28.92 (16.29–46.23)	0.057 **	27.45 (16.29–46.23)	32.88 (21.36–62.27)	0.099
*Laboratory examination*														
FPG (mmol/L) (mean ± SD)	108.50 ± 6.12	109.07 ± 6.22	107.45 ± 5.87	0.334	108.48 ± 4.62	108.56 ± 6.49	0.883	210.20 ± 27.83	213.1 ± 25.63	208.0 ± 29.48	0.049 *	209.03 ± 29.39	212.1 ± 25.45	0.081
2hPG (mmol/L) (mean ± SD)	167.20 ± 14.14	167.97 ± 15.65	166.79 ± 13.45	0.145	166.1 ± 12.02	167.47 ± 14.71	0.001 *	332.30 ± 40.23	339.2 ± 37.93	330.4 ± 42.28	0.146	330.11 ± 41.60	336.13 ± 38.48	0.594
HbA1c (%) (mean ± SD)	5.82 ± 0.52	5.86 ± 0.55	5.72 ± 0.49	0.748	5.80 ± 0.45	5.84 ± 0.52	0.334	10.17 ± 1.98	11.45 ± 1.19	8.81 ± 2.65	0.032 *	8.60 ± 1.62	11.61 ± 1.43	0.024 *
TC (mmol/L) (mean ± SD)	185.30 ± 56.18	188.52 ± 55.22	180.66 ± 43.14	0.262	182.34 ± 45.50	186.67 ± 54.74	0.979	220.20 ± 54.43	231.3 ± 43.46	218.9 ± 40.83	0.015 *	213.9 ± 43.08	229.1 ± 60.92	0.041 *
TG (mmol/L) (mean ± SD)	125.50 ± 65.81	130.91 ± 87.35	122.76 ± 52.68	0.958	112.0 ± 31.71	129.03 ± 71.96	0.129	188.00 ± 108.60	199.4 ± 114.2	180.3 ± 102.3	0.089	181.28 ± 112.12	192.9 ± 104.03	0.166
LDL-C (mmol/L) (mean ± SD)	103.70 ± 42.13	104.21 ± 40.83	102.80 ± 45.30	0.394	101.66 ± 40.56	111.6 ± 47.84	0.159	136.50 ± 48.70	142.2 ± 55.82	129.6 ± 38.38	0.021 *	132.08 ± 53.62	139.57 ± 40.46	0.205
HDL-C (mmol/L) (mean ± SD)	53.64 ± 12.78	50.39 ± 12.18	55.24 ± 12.85	0.220	57.54 ± 12.08	52.61 ± 12.83	0.243	43.28 ± 13.38	40.53 ± 14.71	47.24 ± 11.56	0.496	42.40 ± 12.19	46.65 ± 12.05	0.119
eGFR (mL/min/1.73 m^2^)														
CKD-EPI [median (range)]	87.5 (38.0–117.0)	86.0 (38.0–117.0)	89.0 (45.0–110.0)	0.766	85.0 (61.0–108.0)	88.0 (38.0–117.0)	0.221	90.5 (29.0–129.0)	85.0 (29.0–117.0)	95.0 (45.0–129.0)	0.001 *	95.0 (45.0–129.0)	85.0 (29.0–117.0)	0.005 *
MDRD-STUDY [median (range)]	83.5 (38.0–147.0)	81.0 (55.0–119.0)	88.0 (38.0–147.0)	0.693	82.0 (38.0–142.0)	86.0 (58.0–147.0)	0.805	89.5 (29.0–147.0)	79.0 (29.0–141.0)	95.5 (45.0–147.0)	0.026 *	96.0 (45.0–147.0)	77.0 (29.0–141.0)	0.033 *
BUN (mg/dL) (mean ± SD)	41.62 ± 17.07	42.21 ± 17.40	40.42 ± 16.56	0.788	41.90 ± 19.7	41.78 ± 16.43	0.854	40.79 ± 14.81	37.76 ± 12.94	43.18 ± 14.71	0.170	38.67 ± 13.35	42.15 ± 15.63	0.130
Crea (mg/dL) (mean ± SD)	0.87 ± 0.37	0.88 ± 0.37	0.87 ± 0.38	0.947	0.94 ± 0.43	0.86 ± 0.35	0.660	0.91 ± 0.24	0.87 ± 0.19	0.94 ± 0.28	0.241	0.93 ± 0.27	0.88 ± 0.19	0.170
UA (mg/dL) (mean ± SD)	5.61 ± 1.46	5.29 ± 1.82	4.76 ± 1.36	0.453	5.23 ± 1.60	5.16 ± 1.57	0.526	4.98 ± 1.61	4.76 ± 1.46	5.16 ± 1.71	0.203	5.11 ± 1.70	4.78 ± 1.44	0.507
Hb (g/dL) (mean ± SD)	13.50 ± 1.86	13.24 ± 1.64	13.63 ± 1.96	0.615	13.89 ± 1.61	13.40 ± 1.92	0.157	14.55 ± 1.73	14.72 ± 1.79	13.21 ± 2.68	0.154	14.47 ± 1.68	14.67 ± 1.81	0.185
WBC (×10^3^/μL) (mean ± SD)	7.92 ± 1.93	8.02 ± 1.93	7.32 ± 1.64	0.038 *	7.73 ± 1.95	8.08 ± 1.98	0.141	8.68 ± 2.83	9.25 ± 1.56	8.12 ± 3.58	0.035 *	8.42 ± 3.50	9.13 ± 2.32	0.056 **
NEU (×10^3^/μL) (mean ± SD)	4.87 ± 1.47	5.05 ± 1.46	4.22 ± 1.37	0.018 *	4.70 ± 1.33	4.95 ± 1.54	0.488	5.29 ± 3.02	5.85 ± 2.83	4.93 ± 4.19	0.026 *	5.46 ± 2.99	5.10 ± 3.07	0.141
LYM (×10^3^/μL) (mean ± SD)	2.27 ± 0.71	2.29 ± 0.67	2.24 ± 0.79	0.915	2.26 ± 0.36	2.28 ± 0.78	0.926	2.67 ± 0.89	2.55 ± 0.83	2.77 ± 0.94	0.274	2.52 ± 0.83	2.77 ± 0.93	0.335
MON (×10^3^/μL) (mean ± SD)	0.53 ± 0.16	0.55 ± 0.18	0.52 ± 0.15	0.220	0.52 ± 0.15	0.53 ± 0.16	0.454	0.55 ± 0.18	0.55 ± 0.21	0.56 ± 0.15	0.799	0.55 ± 0.16	0.56 ± 0.21	0.920
PLT (×10^3^/μL) (mean ± SD)	248.30 ± 75.02	264.18 ± 57.98	235.41 ± 82.38	0.066	222.38 ± 66.96	259.54 ± 76.86	0.130	233.10 ± 71.48	218.2 ± 78.12	254.5 ± 66.48	0.085	227.12 ± 64.23	244.59 ± 82.42	0.108
*Malnutrition*														
ALB (g/dL) (mean ± SD)	6.19 ± 0.37	5.74 ± 0.19	6.41 ± 0.18	<0.0001 *	6.30 ± 0.30	6.16 ± 0.38	0.230	3.89 ± 0.78	3.13 ± 0.34	4.49 ± 0.42	<0.0001 *	4.41 ± 0.48	3.07 ± 0.32	<0.0001 *
CRP (mg/dL) (mean ± SD)	27.46 ± 19.65	29.94 ± 20.44	26.24 ± 19.28	0.551	11.77 ± 10.76	31.63 ± 19.25	0.0002 *	47.91 ± 36.27	50.65 ± 37.74	44.42 ± 34.41	0.130	45.87 ± 35.92	49.21 ± 36.73	0.289
ESR (mm/1st h) (mean ± SD)	35.01 ± 25.42	36.31 ± 31.01	34.36 ± 22.35	0.885	34.43 ± 25.96	34.70 ± 24.98	0.976	38.43 ± 16.99	41.91 ± 16.70	35.05 ± 17.36	0.063	36.03 ± 18.91	42.05 ± 14.33	0.092
PNI (mean ± SD)	61.87 ± 3.67	57.44 ± 1.87	64.06 ± 1.93	<0.0001 *	63.06 ± 3.01	61.57 ± 3.78	0.231	38.87 ± 7.80	31.27 ± 3.37	44.86 ± 4.21	<0.0001 *	44.11 ± 4.76	30.70 ± 3.15	<0.0001 *
GPS, *n*														
1	21	3	18		-	-		61	5	56		-	-	
2	79	30	49		-	-		39	39	0		-	-	

SBP: systolic blood pressure; DBP: diastolic blood pressure; WC: waist circumference; HC: hip circumference; WHR: waist-to-hip ratio; WHtR: waist-to-height ratio; BMI: body mass index; BAI: body adiposity index; FPG: fasting plasma glucose; 2hPG: two-hour plasma glucose after a 75 g oral glucose tolerance test; HbA1c: glycosylated hemoglobin A1c; TC: total cholesterol; TG: total triglycerides; LDL-C: low-density lipoprotein cholesterol; HDL-C: high density lipoprotein cholesterol; e-GFR: estimated glomerular filtration rate; CKD-EPI: chronic kidney disease epidemiology collaboration; MDRD-Study: modification of diet in renal disease study; BUN: blood urea nitrogen; CREA: creatinine; UA: uric acid; Hb: hemoglobin; WBC: white blood cells/leukocytes; NEU: neutrophils; LYM: lymphocytes; MON: monocytes; PLT: platelets; ALB: albumin; CRP: C-reactive protein; ESR: erythrocyte sedimentation rate; PNI, prognostic nutritional index; GPS: Glasgow Prognostic Score; SD: standard deviation; * *p* < 0.05: statistically significant; **: reached the significance limit.

**Table 3 diagnostics-14-02661-t003:** Associations of PNI with BMI, WHR, WHtR, and BAI in the preDM and T2DM groups.

Variables (Mean ± SD)	preDM Group (*n* = 100)		T2DM Group (*n* = 100)	
	PNI	*p*-Value from Kruskal–Wallis/ One-Way ANOVA	PNI	*p*-Value from Kruskal–Wallis/ One-Way ANOVA
BMI category (kg/m^2^)				
Normal weight	63.03 ± 3.30	0.019 *	40.44 ± 9.28	0.442
Over weight	61.05 ± 3.96		39.87 ± 8.13	
Obese	60.88 ± 3.49		38.57 ± 7.27	
WHR				
Quarter 1	62.88 ± 3.37	0.059 **	39.89 ± 7.40	0.782
Quarter 2	61.55 ± 4.61		39.16 ± 8.44	
Quarter 3	62.64 ± 3.09		38.66 ± 7.28	
Quarter 4	60.44 ± 3.06		37.80 ± 8.31	
WHtR				
Quarter 1	61.77 ± 3.30	0.845	41.71 ± 7.01	0.109
Quarter 2	61.41 ± 3.49		38.59 ± 8.30	
Quarter 3	62.03 ± 4.66		37.67 ± 7.98	
Quarter 4	62.31 ± 3.20		36.55 ± 7.28	
BAI				
Quarter 1	60.57 ± 3.48	0.053 **	40.21 ± 7.58	0.695
Quarter 2	61.74 ± 4.33		38.77 ± 8.75	
Quarter 3	61.95 ± 3.18		38.74 ± 7.23	
Quarter 4	63.10 ± 3.23		37.79 ± 7.82	

BMI: body mass index; WHR: waist-to-hip ratio; WHtR: waist-to-height ratio; BAI: body adiposity index; * *p* < 0.05: statistically significant; **: reached the significance limit.

## Data Availability

The data used to support the findings of this study are available from the corresponding author upon reasonable request.

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
