# Peer review of "Correlation Between Prognostic Nutritional Index, Glasgow Prognostic Score, and Different Obesity-Related Indices in People with Diabetes or Prediabetes"

_diagnostics, 2024, doi:10.3390/diagnostics14232661_

Round 1
Reviewer 1 Report
Comments and Suggestions for Authors
Dear authors
The manuscript is excellent. The idea is very important and introduced well.
Nutrition and inflammation play important role in diabetes and its complecations. So; studying the correlation between prognostic nutritional index, glasgow prognostic score, and different obesity-related indices in a patients with diabetes or prediabetes is very important.
These highlighted items in the attachment, the authors need to add the units used for determination of BUN, Cre and UA.

Author Response
Dear Reviewer,
Thank you very much for taking the time to analyze our manuscript, and for your kind appreciation and valuable suggestions.
All the typing recommended changes were performed in the body of our manuscript, with the Track Changes function activated.
Comments and Suggestions for Authors
The manuscript is excellent. The idea is very important and introduced well.
Nutrition and inflammation play important role in diabetes and its complecations. So; studying the correlation between prognostic nutritional index, glasgow prognostic score, and different obesity-related indices in a patients with diabetes or prediabetes is very important.
These highlighted items in the attachment, the authors need to add the units used for determination of BUN, Cre and UA.
- We added the units used for the determination of BUN, Cre, and UA

Reviewer 2 Report
Comments and Suggestions for Authors
Comments to the Authors,
I read with great interest the manuscript entitled ” Correlation between Prognostic Nutritional Index, Glasgow Prognostic Score, and Different Obesity-Related Indices in a Patients with Diabetes or Prediabetes".
Title:
Correlation between Prognostic Nutritional Index, Glasgow Prognostic Score, and Different Obesity-Related Indices in a Patients with Diabetes or Prediabetes.
· It's better to avoid using the term patients and replace it with people with.
· Remove the a in a patients.
Abstract:
It's better to avoid using the terms we and our and replace it with a more neutral term e.g. the current study.
You mentioned that "The Enzyme- Linked Immunosorbent Assay (ELISA) method was applied." but in the methodology, non of the assessed parameters was by Elisa. You have to mention the used investigations and the method of each.
Please put a results section with your p values and a conclusion section with the major conclusion.
Methodology:
· It’s advised to do sample size collection and add how the patients were selected.
· In the prediabetes group, yo mention any obese, hypertensive or dyslipidemia person, what is your justification for that. These persons are not essentially prediabetic.
· How did you exclude type 1 diabetes and other forms of diabetes?
· It would have been nice to include a normal group as a control.
· You mention you excluded those with complications and at first you said you only included newly diagnosed ones. Thes people should have no complications.
· Was it a retrospective study or cross sectional one? you mentioned that "after collecting anthropometric data, we brought the subjects to the laboratory for further investigation." I think it's a cross sectional one.
· What about insulin resistance and HOMA IR?
· It would be nice to add data about diabetes therapy, type, dose and compliance.
· It would be nice to add data about other comorbidities.
Results:
· Utilizing PNI cut-off values of 61.0 and 38.0, how did you get these cut off values? Do you have reference for them.
· It would be nice to add a correlation between HbA1c and both indices in both groups.
· It would be nice to add multivariate regression for the factors most associated with T2DM and prediabetes development.

English editing is required.
Author Response
Dear Reviewer,
Thank you very much for taking the time to analyze our manuscript, and for your kind appreciation and valuable suggestions.
All the typing recommended changes were performed in the body of our manuscript, with the Track Changes function activated.
Comments and Suggestions for Authors
I read with great interest the manuscript entitled ” Correlation between Prognostic Nutritional Index, Glasgow Prognostic Score, and Different Obesity-Related Indices in a Patients with Diabetes or Prediabetes".
Title:
Correlation between Prognostic Nutritional Index, Glasgow Prognostic Score, and Different Obesity-Related Indices in a Patients with Diabetes or Prediabetes.
-It's better to avoid using the term patients and replace it with people with.
- Remove the a in a patients.
- Revised according to the recommendations made.
Abstract:
- It's better to avoid using the terms we and our and replace it with a more neutral term e.g. the current study.
- Revised according to the recommendations made.
- You mentioned that "The Enzyme- Linked Immunosorbent Assay (ELISA) method was applied." but in the methodology, non of the assessed parameters was by Elisa.
- Revised according to the recommendations made.
- You have to mention the used investigations and the method of each.
- We mentioned in the Laboratory Investigations Section
Laboratory data, blood urea nitrogen (BUN), creatinine (CREA), uric acid (UA), fasting plasma glucose (FPG), two-hour plasma glucose after a 75-gram oral glucose tolerance test (2hPG), glycosylated hemoglobin A1c (HbA1c), total cholesterol (TC), total triglycerides (TG), low-density lipoprotein cholesterol (LDL-C), high-density lipoprotein cholesterol (HDL-C), C-reactive protein (CRP), and albumin (ALB) were determined using the chemiluminescence immunological technique and an automatic immunoassay analyzer (Cobas e411, Roche Diagnostics GmbH, Mannheim, Germany).
Using flow cytometry and Coulter's principle, we were able to obtain an extended leukocyte formula of 5 diff (Ruby Cell-Dyne, Abbott, Abbott Park, IL, USA) and determine the hemoleucogram markers: hemoglobin (Hb), white blood cells/leukocytes (WBC), neutrophils (NEU), lymphocytes (LYM), monocytes (MON), platelets (PLT), and hemoglobin (Hb).
Measurements of serum creatinine were made, and the Chronic Kidney Dis-ease Epidemiology Collaboration (CKD-EPI) formula [33], and the Modification of Diet in Renal Disease Study (MDRD-Study) [34] was used to determine the estimated glomerular filtration rate (eGFR).
- Please put a results section with your p values and a conclusion section with the major conclusion.
- Revised according to the recommendations made.
Methodology:
- It’s advised to do sample size collection and add how the patients were selected.
- We mentioned in the Materials and Methods Section
In this study, one hundred eighty-five consecutive patients with newly diagnosed T2DM were enrolled, while one hundred patients with preDM who matched the inclusion criteria in terms of age, gender ratio, and urban/rural location made up the control group.
Patients with chronic microvascular complications of T2DM at diagnosis that include diabetic peripheral polyneuropathy, diabetic kidney disease, and diabetic retinopathy were excluded from the study. Diabetic retinopathy (DR) was diagnosed following a dilated fundus examination [31]. As advised by the American Diabetes Association (ADA), diabetic peripheral neuropathy was evaluated using a combination of temperature sensation (for small fiber function) and vibration sensation (for large fiber function) tests, as well as the presence of characteristic symptoms (pain, dysesthesias, numbness) [31]. Guidelines from Kidney Disease: Improving Global Outcomes (KDIGO) were used to assess the existence of CKD [32].
- In the prediabetes group, yo mention any obese, hypertensive or dyslipidemia person, what is your justification for that. These persons are not essentially prediabetic.
- We included these patients who met the criteria mentioned by the ADA in Diagnosis and classification of diabetes: Standards of Care in Diabetes—2024.
- How did you exclude type 1 diabetes and other forms of diabetes?
- We mentioned in the Materials and Methods Section
- Regarding cases of type I diabetes, I removed the specification from the text because we did not have any cases.
- It would have been nice to include a normal group as a control.
- We did not intend to include a group of normal people as a control group.
- You mention you excluded those with complications, and at first you said you only included newly diagnosed ones. These people should have no complications.
- Of all the patients newly diagnosed with T2DMs, after investigation and applying the specified criteria, we included in the study, as specified in the Assessment of diabetes and prediabetes ubsection, only the 100 patients.
- Was it a retrospective study or cross sectional one? you mentioned that "after collecting anthropometric data, we brought the subjects to the laboratory for further investigation." I think it's a cross sectional one.
- Revised : it was a cross sectional one.
- What about insulin resistance and HOMA IR?
- It would be nice to add data about diabetes therapy, type, dose and compliance.
- At the time of diagnosis, the first visit to the hospital, when they were investigated, they did not receive medication.
- If we had proposed that the study be carried out in dynamics, taking into account a second visit, we would have specified data about diabetes therapy, type, dose, and compliance.
- It would be nice to add data about other comorbidities.
- They had no other comorbidities
Results:
- Utilizing PNI cut-off values of 61.0 and 38.0, how did you get these cut off values? Do you have reference for them.
- These were median values ​​obtained by us.
- We mentioned in the Prognostic Nutritional Index and Glasgow Prognostic Score calculations Section
Provided that 61.00 was the median value among the 100 preDM patients, and 38.00 for the 100 T2DM patients, respectively, we used the median of PNI scores as classified criteria and were divided into two groups: Low PNI (<61.00, and <38.00, respectively) group and high PNI (≥61.00, and ≥38.00, re-spectively) group.
- It would be nice to add a correlation between HbA1c and both indices in both groups.
- We mentioned in Correlations between PNI and the obesity-related indexes in the preDM and T2DM groups
Additionally, the PNI values expressed a negative weak correlation with BMI (rho = -0.279, p-value = 0.015), and HbA1c (rho = -0.245, p-value = 0.025).
- It would be nice to add multivariate regression for the factors most associated with T2DM and prediabetes development.

Round 2
Reviewer 2 Report
Comments and Suggestions for Authors
All comments were addressed properly.